# Morphological Changes in the Foveal Avascular Zone after Panretinal Photocoagulation for Diabetic Retinopathy Using OCTA: A Study Focusing on Macular Ischemia

**DOI:** 10.3390/medicina58121797

**Published:** 2022-12-06

**Authors:** Ken Hoshiyama, Takao Hirano, Kazutaka Hirabayashi, Masumi Wakabayashi, Motoharu Tokimitsu, Toshinori Murata

**Affiliations:** Department of Ophthalmology, Shinshu University School of Medicine, Matsumoto 390-8621, Japan

**Keywords:** diabetic retinopathy, diabetic macular ischemia, DMI, panretinal photocoagulation, PRP, optical coherence tomography angiography, OCTA, foveal avascular zone, FAZ

## Abstract

*Background and objectives*: This study aimed to analyze the morphological changes in the foveal avascular zone (FAZ) after panretinal photocoagulation (PRP) in patients with diabetic retinopathy, with a particular focus on the presence or absence of comorbid diabetic macular ischemia (DMI), using optical coherence tomography angiography (OCTA). *Materials and Methods*: Treatment-naïve 25 eyes of 16 patients who received PRP were examined in this retrospective case series. FAZ area, perimeter, and circularity were calculated on a 3 × 3-mm en-face OCTA image before PRP (baseline) and 1 and 3 months after PRP. The patients were divided into two groups according to coexisting DMI, and each group was statistically analyzed. *Results*: In patients with DMI (9 eyes), FAZ area significantly decreased from the baseline to 3 months after PRP (0.86 ± 0.56 to 0.61 ± 0.31 mm^2^, *p* = 0.018), whereas FAZ perimeter and circularity remained unchanged following treatment (*p* = 0.569 and 0.971, respectively). In patients without DMI (16 eyes), FAZ parameters did not show statistically significant changes across the 3-month follow-up period. *Conclusion*: PRP significantly reduces FAZ area in patients with DMI.

## 1. Introduction

Diabetic retinopathy (DR) is a retinal microangiopathy that develops from vascular endothelial damage associated with hyperglycemia. Capillary ischemia induces neovascularization, which progresses to proliferative diabetic retinopathy (PDR), one of the leading causes of blindness in developed countries [1,2]. Panretinal photocoagulation (PRP) is the gold standard for treating DR to prevent visual loss by improving the oxygenation of the ischemic inner retinal layers [3]. In addition, regression of intraretinal microvascular abnormalities (IRMAs) and neovascularization following PRP may normalize the macular blood flow [4]. Several studies have reported that PRP influences the retinal blood flow, causes large vessel constriction, and decreases blood flow [4,5,6,7].

Optical coherence tomography angiography (OCTA) is a noninvasive and repeatable method used to assess detailed morphological and foveal vascular changes [8]. Although previous studies have explored the large vessel effects of PRP, several recent studies have examined the effects of PRP on microvasculature using OCTA and reported changes in OCTA parameters, such as increased macular microvascular density [9,10,11]. In contrast, some prior studies using OCTA have shown that PRP does not alter the foveal avascular zone (FAZ) area [10,11,12].

Diabetic macular ischemia (DMI) is a complication of diabetic retinopathy that results in irreversible vision loss. Traditionally, DMI is defined as a decreased retinal vascular density and an enlargement of the FAZ, as evidenced by fluorescein angiography (FA) with an enlarged area of the FAZ of at least 0.5 mm^2^ or a parafoveal capillary dropout in at least one parafoveal quadrant if the FAZ is <0.5 mm^2^ [13]. OCTA correlates with FA in diagnosing and quantifying DMI [14,15]. We hypothesized that with the more significant enlargement of the FAZ and parafoveal capillary dropout at the start of treatment, the presence of DMI at baseline would likely induce morphological changes in the FAZ after PRP.

This study aimed to analyze the morphological changes of FAZ after PRP in patients with DR using OCTA, focusing on the presence of DMI.

## 2. Materials and Methods

### 2.1. Patient Population

In this retrospective case series study, patients who received PRP treatment in the Department of Ophthalmology, Shinshu University Hospital between April 2018 and October 2019 were followed up at least 3 months after receiving PRP. The Institutional Review Board of Shinshu University approved the study protocol. The whole procedure was performed following the tenets of the Declaration of Helsinki. Owing to the retrospective study design and use of de-identified patient data, the institutional review board waived the requirement for informed consent. Treatment-naïve eyes with severe non-PDR (NPDR) or PDR were eligible. All patients underwent wide-field fundus FA for the staging of the DR before PRP. The exclusion criteria were as follows: (1) patients with a history of previous treatment for diabetic retinopathy, including intravitreal injection of anti-vascular endothelial growth factors, laser therapy, or vitrectomy; (2) patients with retinal diseases other than DR, such as age-related macular degeneration and retinal vein occlusion; and (3) patients with significant media opacity caused by different conditions (e.g., corneal opacity, dense cataract, significant diabetic macular edema, and vitreous hemorrhage) that decreased image quality. The eyes were divided into two groups according to the presence or absence of DMI at the baseline: no DMI group, eyes without DMI; and DMI group, eyes with DMI. The presence or absence of DMI was defined from the OCTA image measurements obtained, as described below. Each group was statistically analyzed separately.

### 2.2. Panretinal Photocoagulation

Photocoagulation was performed through a contact lens (Ocular Mainster PRP 165, Ocular Instruments, Bellevue, WA, USA) using a 577-nm yellow laser (MC-500 Vixi, Nidek, Gamagori, Japan) with a 200-μm spot diameter and a 0.20-s duration. The power of the laser was individually adjusted to produce yellowish-white coagulative spots. PRP was performed by a single retinal specialist, in 4 sessions, with 2-week intervals between each session, according to the recommendations of the Early Treatment Diabetic Retinopathy Study group study [16].

### 2.3. OCTA Imaging

Patients underwent imaging using the swept-source OCTA (SS-OCTA) system (PLEX^®^ Elite 9000, Carl Zeiss Meditec, Dublin, CA, USA). For each eye, a 3 × 3-mm OCTA image was acquired. Poor quality images with either significant motion artifacts or extensive incorrect segmentation were excluded and repeated. To calculate quantitative FAZ metrics on the whole retina slab, we used Macular Density version 0.7.2 on the ARI Network (Zeiss Portal version 5.4–1206). FAZ parameters were assessed as area, perimeter, and circularity. DMI was defined as a FAZ area on OCTA of ≥0.5 mm^2^ or a FAZ area < 0.5 mm^2^ with parafoveal capillary dropout in at least one quadrant, as previously reported [14,15]. Furthermore, circularity was described as a uniformity index with a value of 1 representing a perfect circle. OCTA examination was conducted before, 1 month after, and 3 months after PRP (Figure 1).

### 2.4. Statistical Analyses

Statistical analyses were performed using GraphPad Prism version 9.4.1 (GraphPad Inc., La Jolla, CA, USA). Continuous variables are represented as mean ± standard deviation and analyzed using the Friedman test for three consecutive measurements. If a significant difference was detected, the successive binary differences between values were analyzed using the Wilcoxon signed-rank test. A *p* value < 0.05 was considered a statistically significant difference.

## 3. Results

A total of 25 treatment-naïve eyes of 21 patients, including 16 males and 5 females, were examined in this study. The mean age was 57.1 ± 10.6 (range, 34–76) years. The baseline DR severities of 19 (76%) and 6 (24%) eyes were severe NPDR and PDR, respectively. The demographic and clinical data of the patients are shown in Table 1. There were no significant differences between the demographic characteristics of the no DMI (16 eyes) and DMI (9 eyes) groups, except for the FAZ area (0.31 ± 0.11 mm^2^ for the no DMI group vs. 0.86 ± 0.56 mm^2^ for the DMI group, *p* < 0.001). Overall, study patients had increased central macular thickness (CMT) and worsened best corrected visual acuity (BCVA) after PRP, but FAZ area, perimeter, and roundness did not change significantly (Table 2).

The results obtained from the no DMI group are summarized in Table 3.

In the no DMI group, FAZ area, perimeter, and circularity at baseline were 0.31 ± 0.11 mm^2^, 2.46 ± 0.56 mm, and 0.65 ± 0.11, respectively; all these FAZ parameters did not show statistically significant changes across the 3-month follow-up period. CMT significantly increased from 310.8 ± 60.4 µm at baseline to 318.8 ± 44.9 µm at 1 month (*p* = 0.027) and 338.0 ± 76.9 µm at 3 months after PRP (*p* = 0.010), whereas BCVA did not change over the observation period.

The results obtained from the DMI group are summarized in Table 4.

In the DMI group, no significant changes were observed in BCVA and CMT. Univariate analysis showed a statistically significant difference in FAZ area among the three points (*p* = 0.047). The FAZ areas were lower at 1 and 3 months post-PRP compared to baseline, but with no statistically significant difference at 1 month (0.86 ± 0.56, 0.61 ± 0.31, and 0.62 ± 0.37 mm^2^; *p* = 0.059 and *p* = 0.018, respectively). No significant difference was observed for FAZ perimeter and circularity between consecutive measurements. The representative case of the DMI group is shown in Figure 2.

## 4. Discussion

In this study, SS-OCTA was used to observe the morphological changes in the FAZ after PRP in patients with DR. In the DMI group, a significant decrease in FAZ area was observed 3 months after PRP. In contrast, in the no DMI group, no substantial changes in FAZ parameters were observed after PRP.

A number of prior studies have shown decreased retinal blood flow in eyes with DR treated with PRP using various devices [4,5,6,7,17]. For example, in 1982, Feke et al. investigated retinal blood flow changes after PRP using the laser Doppler technique. They reported decreased flow pulsatility and retinal arterial and venous diameters [5]. Similar vascular changes in the large vessel after PRP have been demonstrated by other techniques, such as Doppler optical coherence tomography and laser speckle flowgraphy [7,17].

It has been hypothesized that PRP increases oxygen flow to the inner retina by destroying oxygen-consuming photoreceptor cells on the outer retina, thereby decreasing the stimulus for angiogenesis [18,19]. In addition, increased oxygenation to the inner retina decreases the release of nitric oxide from the vascular endothelium with an overall decreased retinal vessel diameter [20].

OCTA is a noninvasive and repeatable method used to assess detailed morphological and foveal vascular changes [8,21,22,23]. Therefore, it is helpful for the follow-up of foveal microvascular changes after PRP. This finding has not been reported in previous studies, which are limited to studies of large vessel hemodynamics.

A previous report using OCTA to evaluate microvascular parameters in patients with high-risk PDR before, 1 month after, and 3–6 months after PRP suggested an overall redistribution of blood flow to the posterior pole after PRP, in line with their electrical circuit model of retinal circulation that mathematically studied the hemodynamics of the retinal vasculature network [24]. Similarly, another study of patients with high-risk PDR, in which vessel density, choroidal flow, and FAZ area were measured before PRP and at 1 and 6 months after PRP, reported statistically significant increases in vessel density at the 1- and 6-month follow-up periods [9]. These increases in vessel density were believed to be due to the closure of peripheral neovascular and IRMAs following PRP, which redistributed blood flow to the macula.

The increased blood flow in the posterior pole after PRP identified in OCTA may affect FAZ. Previous studies have observed morphological changes in the FAZ after PRP, but the results are inconsistent.

Some reports have revealed the FAZ area was constricted and became significantly more circular after PRP in patients with high-risk PDR [9,25]. In the study, including patients with NPDR and PDR, the FAZ area decreased slightly 12 months after PRP, although the decrease was not statistically significant (*p* = 0.551) [10]. A possible underlying mechanism for FAZ morphological changes after PRP can be the re-establishment of macular microvasculature from regression of peripheral neovascularization or IRMAs [26,27].

In this study, which included patients with NPDR and PDR, there were no significant changes in FAZ parameters after PRP in all study patients and the no DMI group, despite a significantly smaller FAZ area in the DMI group 3 months after PRP.

One study evaluating FAZ changes after PRP reported that the FAZ area was not significantly affected by PRP at 1 and 6 months after PRP. Although the subjects of their study were patients with PDR, the baseline FAZ area (0.33 ± 0.19 mm^2^) was as small as in the no DMI group in our study (0.31 ± 0.11 mm^2^) [12]. In contrast, a decrease in FAZ area and an increase in FAZ circularity were observed 1 and 6 months after PRP in patients with NPDR with a large FAZ area (> 0.35 mm^2^) [28].

Previous studies using OCTA have demonstrated that FAZ enlargement is more common in patients with diabetes compared to normal healthy patients, regardless of the presence or absence of concomitant diabetic retinopathy [29]. As the severity of DR increases, the FAZ becomes more prominent and irregular in shape [30]. Differences in the severity of DR and the size of the FAZ area may be the reason for the different results in these studies on FAZ changes after PRP. The duration of observation, differences in OCTA device used for measurement, and irradiation method of PRP may also affect the results. However, the limited number of subjects precluded us from performing subgroup analysis to evaluate this theory further.

In the present study, there was a significant reduction in FAZ area after PRP in the DMI group, but there was no improvement in visual acuity. There are several possible reasons for this. First, the baseline BCVA for the DMI group included in this study was relatively good at 0.05 ± 0.19. This may have influenced the no statistically significant improvement in visual acuity due to the ceiling effect. Second, the observation period may have been significantly short to achieve visual acuity improvement. Third, there may be no association between the morphological change of reduced FAZ area and improved visual acuity. As shown in Figure 2, the reduction in the FAZ area after PRP is due to local capillary revascularization within the FAZ. Further studies using microperimetry, which can assess local retinal sensitivity, are required to determine whether this capillary revascularization improves retinal sensitivity in the dominant region.

This study has some limitations. The major limitation of this study is the limited number of patients. Furthermore, to ensure the quality of OCTA images, this study did not include patients with poor visual acuity, which can cause poor fixation or patients with severe PDR. To investigate a definite association between PRP and FAZ parameters, it would have been ideal to include more patients with severe DR and DMI for analysis and for analyzing them in groups by severity. However, the sample sizes would have been small. Another significant limitation is that blood glucose levels were not considered. Reasonable glycemic control is known to prevent the development or progression of microvascular complications of diabetes [31], and blood glucose trends may affect the morphological changes of FAZ. An additional limitation is the short observation period. Although the technology to visualize retinal vessels by OCTA is evolving rapidly, it remains difficult to repeatedly obtain good-quality images from any patient with DR. Further innovations in OCTA devices will be useful in conducting large-scale studies with a more significant number of patients and long-term observation.

## 5. Conclusions

Our study suggests that PRP significantly reduces FAZ area in patients with DMI following PRP.

## Figures and Tables

**Figure 1 medicina-58-01797-f001:**
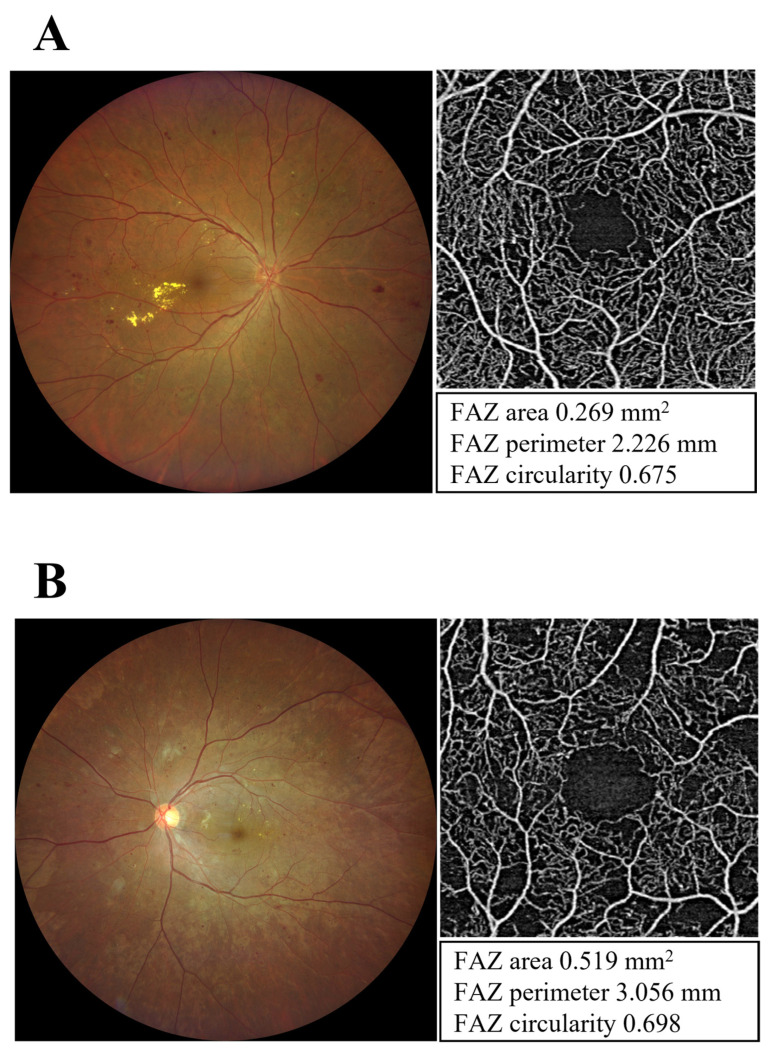
Fundus photograph of diabetic retinopathy and foveal avascular zone (FAZ) on optical coherence tomography angiography (OCTA). FAZ area, perimeter, and circularity were quantitatively evaluated from 3 × 3-mm en-face OCTA images using Macular Density version 0.7.2 on the ARI Network (Zeiss Portal version 5.4-1206) from 3 × 3-mm en-face OCTA images. The eyes were divided into two groups according to the presence or absence of diabetic macular ischemia (DMI) at the baseline: (**A**) no DMI group, eyes without DMI; and (**B**) DMI group, eyes with DMI. DMI was defined as a FAZ area on OCTA of ≥0.5 mm^2^ or a FAZ area < 0.5 mm^2^ with parafoveal capillary dropout in at least one quadrant.

**Figure 2 medicina-58-01797-f002:**
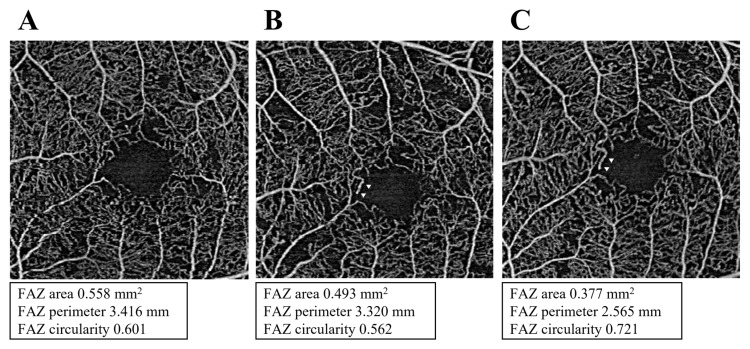
Representative case of the diabetic macular ischemia (DMI) group. (**A**) At baseline, the eye had a large foveal avascular zone (FAZ) with a FAZ area ≥ 0.5 mm^2^, which was diagnosed as DMI. FAZ area was reduced due to local capillary revascularization (white arrowhead) within the FAZ at 1 (**B**) and 3 (**C**) months following panretinal photocoagulation.

**Table 1 medicina-58-01797-t001:** Characteristics of the study group.

Characteristics	All Eyes	No DMI Group	DMI Group	*p* Value
Eyes, *n*	25	16	9	
Age, years, mean ± SD	57.1 ± 10.6	56.6 ± 9.6	56.9 ± 12.5	0.587 ^a^
Sex, male/female	16/5	13/3	5/4	0.169 ^b^
DR severity (severe NPDR/PDR), *n*	19/6	12/4	7/2	0.876 ^b^
HbA1c, %, mean ± SD	8.8 ± 2.2	9.1 ± 2.2	8.5 ± 2.7	0.308 ^a^
BMI, kg/m^2^, mean ± SD	26.3 ± 3.4	25.6 ± 3.4	27.6 ± 3.0	0.081 ^a^
BCVA, logMAR, mean ± SD	0.07 ± 0.29	0.08 ± 0.34	0.05 ± 0.19	0.723 ^a^
CMT, µm, mean ± SD	295.2 ± 62.5	310.8 ± 60.4	267.7 ± 59.6	0.093 ^a^
FAZ area (mm^2^)	0.51 ± 0.43	0.31 ± 0.11	0.86 ± 0.56	<0.001 ^a^

DMI = diabetic macular ischemia; NPDR = non proliferative diabetic retinopathy; PDR = proliferative diabetic retinopathy; BMI = body mass index; BCVA = best corrected visual acuity; logMAR = logarithm of the minimum angle of resolution; CMT = central macular thickness; FAZ = foveal avascular zone; SD = standard deviation. ^a^ Mann–Whitney U test; ^b^ Chi-squared test.

**Table 2 medicina-58-01797-t002:** Summary of results for all study patients and comparison of the baseline and post-panretinal photocoagulation measurements.

Parameters	Baseline	1 Month	3 Months	Friedman Test	Pairwise Comparison	
				(*p* Value)	Baseline vs. 1 Month	Baseline vs. 3 Months
BCVA (LogMAR)	0.07 ± 0.29	0.10 ± 0.27	0.08 ± 0.28	0.036	0.048	0.104
CMT (µm)	295.2 ± 62.5	310.1 ± 48.9	324.4 ± 72.1	0.002	0.009	<0.001
FAZ area (mm^2^)	0.51 ± 0.43	0.44 ± 0.25	0.44 ± 0.27	0.432	-	-
FAZ perimeter (mm)	3.37 ± 2.14	3.01 ± 1.08	3.06 ± 1.22	0.756	-	-
FAZ circularity	0.59 ± 0.15	0.59 ± 0.13	0.59 ± 0.13	0.887	-	-

Mean ± standard deviation; *p* < 0.05 was considered statistically different; BCVA = best corrected visual acuity; logMAR = logarithm of the minimum angle of resolution; CMT = central macular thickness; FAZ = foveal avascular zone in whole retina.

**Table 3 medicina-58-01797-t003:** Summary of the no DMI group results and comparison of the baseline and post-panretinal photocoagulation measurements.

Parameters	Baseline	1 Month	3 Months	Friedman Test	Pairwise Comparison	
				(*p* Value)	Baseline vs. 1 Month	Baseline vs. 3 Months
BCVA (LogMAR)	0.08 ± 0.34	0.10 ± 0.31	0.09 ± 0.32	0.125	-	-
CMT (µm)	310.8 ± 60.4	318.8 ± 44.9	338.0 ± 76.9	0.019	0.027	0.010
FAZ area (mm^2^)	0.31 ± 0.11	0.35 ± 0.14	0.34 ± 0.13	0.377	-	-
FAZ perimeter (mm)	2.46 ± 0.56	2.69 ± 0.65	2.67 ± 0.69	0.452	-	-
FAZ circularity	0.65 ± 0.11	0.61 ± 0.13	0.61 ± 0.13	0.313	-	-

Mean ± standard deviation; *p* < 0.05 was considered statistically different; BCVA = best corrected visual acuity; logMAR = logarithm of the minimum angle of resolution; CMT = central macular thickness; FAZ = foveal avascular zone in whole retina.

**Table 4 medicina-58-01797-t004:** Summary of the DMI group results and comparison of the baseline and post-panretinal photocoagulation measurements.

Parameters	Baseline	1 Month	3 Months	Friedman Test	Pairwise Comparison	
				(*p* Value)	Baseline vs. 1 Month	Baseline vs. 3 Months
BCVA (LogMAR)	0.05 ± 0.19	0.11 ± 0.21	0.08 ± 0.23	0.301	-	-
CMT (µm)	267.7 ± 59.6	294.7 ± 54.5	300.3 ± 58.9	0.107	-	-
FAZ area (mm^2^)	0.86 ± 0.56	0.61 ± 0.31	0.62 ± 0.37	0.047	0.059	0.018
FAZ perimeter (mm)	4.99 ± 2.92	3.73 ± 1.39	3.74 ± 1.66	0.569	-	-
FAZ circularity	0.51 ± 0.18	0.56 ± 0.12	0.58 ± 0.13	0.971	-	-

Mean ± standard deviation; *p* < 0.05 was considered statistically different; BCVA = best corrected visual acuity; logMAR = logarithm of the minimum angle of resolution; CMT = central macular thickness; FAZ = foveal avascular zone in whole retina.

## Data Availability

The data presented in this study are available upon request from the corresponding author. The data are not publicly available due to ethical reasons.

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
