# Peer review of "Morphological Changes in the Foveal Avascular Zone after Panretinal Photocoagulation for Diabetic Retinopathy Using OCTA: A Study Focusing on Macular Ischemia"

_medicina, 2022, doi:10.3390/medicina58121797_

Round 1
Reviewer 1 Report
The authors have conducted a retrospective study on the impact of PRP on microvascular changes in macula area in DR
1- one of the sources for bias which has not been mentioned is the level of blood glucose control which has a well-known effect on the progression of DR.
2- follow up period is short, which is another limitation for the study
3- there is no mention of clinical significance of the results, in my opinion, if the follow up was longer, along with taking into account the blood glucose level control, this issue could be addressed .
Author Response
Please see the attachment.
Reviewer 1
Comment #1. One of the sources for bias which has not been mentioned is the level of blood glucose control which has a well-known effect on the progression of DR.
Response: Thank you for your very important point. We agree that failure to consider the blood glucose control is a limitation of this study. We have added the following statement to the Discussion and included a reference.
Another significant limitation is that blood glucose levels were not considered. Reasonable glycemic control is known to prevent the development or progression of microvascular complications of diabetes [31], and blood glucose trends may affect the morphological changes of FAZ.
- Nathan, D.M.; Genuth, S.; Lachin, J.; Cleary, P.; Crofford, O.; Davis, M.; Rand, L.; Siebert, C. The effect of intensive treatment of diabetes on the development and progression of long-term complications in insulin-dependent diabetes mellitus. N Engl J Med 1993, 329, 977-986, doi:10.1056/nejm199309303291401.
Comment #2. Follow up period is short, which is another limitation for the study.
Response: Thank you for pointing this out. We have mentioned the short follow-up period in the Discussion section.
Comment #3. There is no mention of clinical significance of the results, in my opinion, if the follow up was longer, along with taking into account the blood glucose level control, this issue could be addressed.
Response: We completely agree with your comments. Unfortunately, blood glucose was not measured at all observation points; thus, we have mentioned blood glucose assessment and observation period in the Limitations paragraph.

Reviewer 2 Report
The subject of the paper is very interesting. It evaluated the diabetic macular ischemia after panretinal laser photocoagulation by using a novel tool, OCT- angiography. There are limitted information about this topic, and any aditional evidences are usefull and interesting to the readers.
The main limitation of the study is the limitted number if patients included in the study, which limit the statistical value of the results.
However, the article is well structured and in my opinion, it deserves publications. The discussions could be developed and the refference list updated accordingly.
If the authors may provide images of the retina in association with oct-a images, it would be interesting for the readers to understand how these methods of evaluation complement each other.
Author Response
Please see the attachment.
Reviewer 2
Comment #1. The main limitation of the study is the limitted number if patients included in the study, which limit the statistical value of the results.
Response: We completely agree with this comment. A large-scale study with more participants will be important in the future. Thank you for your valuable comment.
Comment #2. If the authors may provide images of the retina in association with oct-a images, it would be interesting for the readers to understand how these methods of evaluation complement each other.
Response: Thank you for pointing this out. We have presented the fundus photograph side by side with the OCTA image in Figure 1. Thus, we are able to demonstrate the difference in retinal findings with and without diabetic macular ischemia
